# (Cd,Mn)S in the Composite Photocatalyst: Zinc Blende and Wurtzite Particles or Integrowth of These Two Modifications?

**DOI:** 10.3390/ma16020692

**Published:** 2023-01-10

**Authors:** Svetlana Cherepanova, Ekaterina Kozlova

**Affiliations:** Boreskov Institute of Catalysis, 630090 Novosibirsk, Russia

**Keywords:** cadmium-manganese sulfide, X-ray diffraction, disordered structure, stacking faults, anisotropic particle shape

## Abstract

In this study, the crystalline structure and particle shape of Cd_1−x_Mn_x_S (x~0.3) in the composite photocatalysts prepared by hydrothermal synthesis at different temperatures (T = 80, 100, 120, and 140 °C) were analyzed. Along with mixed Cd–Mn sulfide, the catalysts contain a small amount of β-Mn_3_O_4_. XRD patterns of (Cd,Mn)S have features inherent to both cubic zinc blende and hexagonal wurtzite structure. Moreover, XRD peaks are anisotropically broadened. First, the heterogeneous (or two-phased) model was considered by the commonly used Rietveld method. Phase ratio, average crystallite sizes, and strains for both phases were formally determined. However, it was shown that this model is not correct because relatively narrow and broad peaks cannot be fitted simultaneously. Then, the homogeneous model was tested by Debye Function Analysis. This model assumes that particles are statistically homogeneous, but each particle contains lamellar intergrowth of zinc blende and wurtzite modifications. The probability of stacking faults, as well as the average radii of spherical and ellipsoidal particles, were varied. It was shown that nanocrystalline Cd_0.7_Mn_0.3_S particles have an ellipsoidal shape. Ellipsoids are elongated along the direction normal to the plane of defects. An increase in the hydrothermal synthesis temperature from 80 °C to 140 °C leads to an enlargement of particles and a gradual decrease in the probability of stacking faults in the wurtzite structure from 0.47 to 0.36. Therefore, with increasing temperature, the structure of (Cd,Mn)S nanoparticles transforms from almost random polytype cubic/hexagonal (ZB:WZ = 47:53) to a preferably hexagonal structure (ZB:WZ = 36:64). Mn^2+^ ions facilitate CdS phase transformation from zinc blende to wurtzite structure. There is no direct correlation between the structure and photocatalytic activity.

## 1. Introduction

Semiconductor nanocrystals are of great interest for both basic research and practical applications. This is due to their unique size-dependent optical and electronic properties. Semiconductor nanoparticles can also be effective for photocatalytic and photoelectrochemical water-splitting [1]. Among the semiconductors with a narrow band gap energy, nanocrystalline CdS (Eg = 2.4 eV) is the most attractive due to the suitable position of the band edges for the direct splitting of water into hydrogen and oxygen by solar energy [2,3]. Unfortunately, this semiconductor is not stable in aqueous solutions under irradiation [4,5].

The reviews [6,7] consider the most frequently employed strategies for enhancing the efficiency of the metal sulfide photocatalysts. One of the ways is metal ion doping with the creation of mixed metal sulfides (substitutional solid solutions) [6]. The addition of wider-gap semiconductors, such as ZnS and MnS, CdS makes it possible to vary the band gap of the sample, the position of the valence band, and the conduction band. Unlike mixed Cd–Zn sulfides, which can be synthesized by coprecipitation, hydrothermal treatment is usually required to produce mixed Cd–Mn sulfides. Cadmium–zinc sulfides are the widely studied photocatalysts [8]. As for cadmium–manganese sulfides, it was recently shown that they also exhibit high photocatalytic activity [9,10,11,12,13]. Unlike zinc cations, Mn^2+^ cations can be easily oxidized by photogenerated holes to the oxidation state of 3+; thus, the mechanism of the action of (Cd,Zn)S and (Cd,Mn)S photocatalysts may differ. The photocatalytic activity of sulfide photocatalysts strongly depends on their phase composition and crystal structure [14]. Thus, accurate structural analysis of the synthesized samples is very important for the obtainment of active photocatalysts.

Under ambient conditions, bulk cadmium sulfide is crystallized into α-CdS and β-CdS with wurtzite (WZ) and zinc blende (ZB) structures, respectively [15]. The WZ and ZB structures may be considered to be derived from the hexagonal and cubic closest packing of S^2−^ ions, respectively. Cd^2+^ ions occupy half of the tetrahedral holes, and the structures differ only in the packing layers of tetrahedrons. The hexagonal structure is slightly more stable with ∆E_W-ZB_ = −1.1 meV/atom [16].

There are three crystalline modifications of manganese sulfide: α-MnS (rocksalt structure), β-MnS (ZB structure), and γ-MnS (WZ structure) [17]. The β-MnS and γ-MnS modifications are metastable and, when heated to 200–300 °C, become α-MnS. CdS and MnS form Cd_1−x_Mn_x_S with a WZ structure at x ≤ 0.5 [18,19] and a mixture of Cd_1−x_Mn_x_S and α-MnS at x > 0.5 [18].

As for nanocrystalline CdS, there is considerable disagreement about its structure in the literature [20]. Usually, X-ray diffraction (XRD) patterns contain three broad peaks with interplanar distances of d = 3.36, 2.06, and 1.76 Ǻ (near 2θ~27°, 44°, and 52°, CuKα, λ = 1.5418 Ǻ), which are common for both modifications of CdS. However, they cannot be attributed to cubic or hexagonal CdS [21] because of the discrepancy between the relative intensities of experimental diffraction peaks and referenced or simulated XRD patterns. Moreover, there is little or no evidence of either the cubic 200 peak or the hexagonal 102 peak. A main peak around 2θ = 27° tends to exhibit asymmetry or shoulders in the positions of the 100 and 101 peaks in hexagonal CdS. If shoulders are absent, the nanocrystalline CdS is usually interpreted as cubic [22,23,24]. On the contrary, if shoulders are observed, CdS is interpreted as hexagonal [24,25] or a mixture of the cubic and hexagonal forms [26,27,28,29,30,31].

In previous works [21,31,32,33], authors have stated that the structure of CdS nanocrystals can be identified as a random polytype cubic/hexagonal structure, consisting of nearly random stacking sequences of hexagonal planes that form the basis for both the cubic and hexagonal modifications. Such a structure is considered to be an intermediate state of the phase transition from the ZB to the WZ structure [31]. For (Cd,Mn)S nanoparticles (Mn:Cd < 1), XRD patterns are very similar to those of pure CdS; thus, the problems of interpretation remain the same.

Therefore, it is clear that ideal cubic and hexagonal structures of CdS and (Cd,Mn)S do not give a good fit for the experimental data. As for the mixture of phases, it remains unclear why under the same conditions, two modifications are formed simultaneously. A random polytypic structure can arise due to a high concentration of stacking faults (SFs) in the sequence of layers of tetrahedrons in ZB or WZ structures. The appearance of SFs in a ZB structure means the formation of thin lamellar fragments of WZ structure, and vice versa. However, along with equal proportions of hexagonal and cubic structures within the same particle, different proportions of them are also possible.

Recently, for nanocrystalline CdS [34,35], as well as some other compounds [34,35,36,37,38,39], which are also crystallized into ZB and WZ structures, the probability of SFs was determined by the calculation of XRD patterns using the Debye Function Analysis (DFA) [40]. In this method, (1) a model of the nanoparticle taking into account its shape, size, and probability of SFs is generated; (2) the XRD pattern is calculated based on the model with the use of the Debye equation. The DFA method is rarely used in comparison with the Rietveld one because the definition of models of nanocrystals of a definite shape containing SFs is more difficult relative to the definition of perfect crystals. Moreover, additional optimization of model parameters is necessary.

Some authors consider the effect of the structure on the photocatalytic efficiency of CdS-based photocatalysts. Due to the similar band gaps for ZB CdS (2.36 eV) and WZ CdS (2.32 eV), their light absorption capacities are nearly the same. However, the photocatalytic activity of WZ CdS is obviously superior to that of ZB CdS [41,42]. In [43], it was reported that WZ CdS has a stronger ionic character and, consequently, higher charge carrier mobility, which results in a higher photocatalytic activity compared to ZB CdS.

In this work, we analyzed the structure and morphology of Cd_1−x_Mn_x_S (x~0.3) nanoparticles in composite photocatalysts, which are active during the decomposition of water to produce hydrogen [12,13]. The investigation was made by X-ray diffraction via the Debye Function Analysis. The optimized model parameters of nanosized particles synthesized at different temperatures of hydrothermal treatment were compared with the results obtained earlier for nanocrystalline CdS prepared under the same conditions [44].

## 2. Materials and Methods

### 2.1. Sample Synthesis

The synthesis of (Cd,Mn)S-based photocatalysts was carried out as follows: 50 mL of a 0.1 M NaOH solution was added to 100 mL of a mixture of 0.1 M solutions of cadmium chloride and manganese nitrate (Mn:Cd = 0.4:0.6), and the mixture was stirred for 15 min. Then, an excess of 0.1 M Na2S solution was added, and the mixture was stirred for 60 min. The resulting suspension was placed in a Teflon beaker and centrifuged, and the precipitate was washed with distilled water. The precipitate was decanted and left in an oven at room temperature. Then, the samples were hydrothermally treated in water in an autoclave at temperatures of 80, 100, 120, and 140 °C for 24 h. CdS photocatalysts were synthesized in the same way but without the addition of manganese nitrate.

### 2.2. Sample Characterization

#### 2.2.1. Data Collection

The X-ray diffraction experiment was performed with a D8 Advance diffractometer (Bruker, Karlsruhe, Germany) with Cu Kα radiation by scanning over a 2θ region 15 ÷ 90°, with a step of 0.05°. Samples’ morphology was studied using a transmission electron microscope JEM-2010 (JEOL, Akishima, Japan).

#### 2.2.2. Photocatalytic Tests

The photocatalytic activity of the synthesized samples was determined in the reaction of hydrogen evolution under visible light radiation with a wavelength of 450 nm. A reaction suspension containing 100 mL of 0.1M Na2S/0.1M Na2SO3 and 50 mg of the photocatalyst was placed in the reactor and sonicated for 15 min. During all experiments, the reactor was preliminarily purged with an inert gas (argon) to remove oxygen. Then, the reactor was irradiated with a LED light source (0.33 A, 30 W, 45 mW/cm^2^); the reaction suspension was continuously stirred using a magnetic stirrer. For the quantitative determination of the evolved hydrogen, a chromatograph Khromas GKh-1000 (Chromos, Dzerzhinsk, Russia) with a NaX zeolite column was used; argon was used as a carrier gas.

#### 2.2.3. Analysis of XRD Data

Rietveld refinement was carried out with the use of TOPAS (Total Pattern Analysis System, Bruker, Karlsruhe, Germany). Variable parameters were the weight ratio of hexagonal and cubic modifications of (Cd,Mn)S, and average crystallite sizes and strains for both phases.

The Debye calculation of XRD patterns and the optimization of models of nanoparticles of different shapes and sizes containing stacking faults were carried out using the DISCUS program [37], which allows the calculation of the scattering intensity at any point in reciprocal space using the Debye formula [40]:$$I\left(s\right)=\overset{N}{\underset{i}{\sum }}f_{i}{\left(s\right)}^{2}+\overset{N}{\underset{j>i=1}{\sum }}f_{i}\left(s\right)f_{j}\left(s\right)\times \frac{\sin\left({2\pi \mathrm{sr}}_{\mathrm{ij}}\right)}{{2\pi \mathrm{sr}}_{\mathrm{ij}}},$$
where I(s) is the intensity at s = 2sin(θ)/λ for X-rays of wavelength λ diffracted through angle θ, f_i_(s) and f_j_(s) are scattering factors of atoms i and j, and r_ij_ is the distance between atoms i and j. In the first stage, a nanoscale particle of a given shape and size was generated, with the particle-containing SFs. For this, the coordinates of the atoms in the unit cell were defined, and then, this cell was multiplied by translation vectors in two directions into a layer of specified sizes, exceeding the required dimensions. After that, the layers were stacked along the direction normal to the layers to form a particle with a given probability of SFs. Atoms falling within the defined volume threshold of a given size were selected, and the intensities were calculated using the Debye equation. There were 100 such particles generated since each particle has its own distribution of SFs. The intensities for all particles were averaged by summation. Then, the R-factor was calculated, which showed how the simulated XRD pattern matched the experimental one.

Optimized parameters are the probability of SFs α as well as the radii of ellipsoidal particles Rab (in the ab plane) and Rc (along the c direction). The optimization of the models was carried out using a genetic algorithm that changed the value of the optimized parameters in a specified region. For the next generation, structural models are selected with the R-factor values lower relative to the values for the previous generation. Optimization was continued until a good agreement with experimental data was achieved.

#### 2.2.4. Stacking Faults Model

A crystallographic data of both modifications of cadmium sulfide can be found in the Inorganic Crystal Structure Database: α-CdS with WZ structure (CC = 31,074, P63mc) and β-CdS with ZB structure (CC = 31,075, F$\bar{4}$3m). The ZB structure (Figure 1, left) consists of two interpenetrating face-centered cubic (FCC) sublattices of cations and anions. The WZ structure (Figure 1, right) consists of two interpenetrating hexagonal close-packed (HCP) sublattices separated by a translation of about 0.375 along the c direction. These structures result in two different stacking sequences of anions: ABCABC… along the [111] direction for ZB and ABAB… along the c-axis for WZ. The cations are in the tetrahedral sites, so they are in the same positions as the apical anions. The structures of WZ and ZB can be written as aAbBaAbB… and aAbBcCaAbBcC…, respectively. Let us consider how the sequence of tetrahedron layers in the ZB and WZ structures, as well as in these structures containing stacking faults (SF), can be modeled. Suppose that the positions of anions (cations) along the x- and y-axes are fixed as A(a) = (0,0), B(b) = (1/3,2/3), and C(c) = (2/3,1/3).

The ZB structure (Figure 1, left) is formed by successive displacement of the Ab layer by the vector [−1/3, 1/3], which gives tetrahedron layer sequence AbBcCaAbBcCa… (or abbreviated as ABCABC… for anions).

The WZ structure (Figure 1, right) can be defined by alternating two types of layers: Ab (layer 1) and Ac (layer 2). The Ac layer needs to be shifted by the vector [1/3, −1/3] (vector 1) to obtain the Ba layer, and the Ab layer should be shifted by the vector [−1/3, 1/3] (vector 2) relative to the Ba layer. In this case, we obtain a sequence of layers of the AbBaAbBa… type (ABAB…). If we swap the first and the second types of layers in places (with their vectors), we obtain a hexagonal sequence such as AcCaAcCa… (ACAC…).

SFs in the WZ structure are formed when an error occurs in the way that the layers are superposed onto each other. This means that with a certain probability α, the Ab (1) layer is followed by the Ab (1) layer, shifted by the vector [1/3, −1/3] (1), or the Ac (2) layer is followed by the Ac (2) layer, shifted by the vector [−1/3, 1/3] (2). Then, fragments of the ZB structure (AbBc or AcCb) are obtained in the WZ structure AbBaAbBa… or AcCaAcCa… If the probability of stacking faults α is changed from 0 to 1, then the WZ structure accumulating SFs gradually transforms into a ZB structure. One can also vary the probability of SFs in the ZB structure. Consequently, the probability of SFs in ZB will be related to the probability of SFs in WZ as (1 − α).

## 3. Results and Discussion

### 3.1. Rietveld Analysis

Figure 2 shows the experimental X-ray diffraction patterns for (Cd,Mn)S-based photocatalysts prepared by hydrothermal synthesis (HTS) at temperatures (T) of 80, 100, 120, and 140 °C. XRD patterns have features inherent to both cubic ZB structure and hexagonal WZ structure. One can see that the experimental peaks are shifted towards the higher angles relative to the positions of CdS peaks. The shift of the peaks is explained by the fact that the ionic radius of Mn^2+^ in a fourfold coordination (0.66 Å) is smaller than the ionic radius of Cd^2+^ (0.78 Å). According to Vegard’s law, solid solutions Cd_1−x_Mn_x_S (x~0.3) were formed. Additionally, the X-ray diffraction patterns exhibited small peaks of β-Mn_3_O_4_ (hausmanite).

It was shown that Cd_1−x_Mn_x_S (x ≤ 0.4) nanoparticles have both WZ and ZB structures [45]. Therefore, the two-phase model was explored first. To determine the ratio of the cubic and hexagonal modifications of (Cd,Mn)S, we carried out the Rietveld analysis (Figure 3). Calculations showed that at temperatures of 80, 100, 120, and 140 °C, the content of the hexagonal phase was 19%, 24%, 29%, and 45%, respectively, i.e., the amount of WZ-like phase increased with the temperature. However, it should be noted that a peculiarity of this refinement is the extremely large particle sizes (>100 nm) and strains (>1). This is because the high angle peaks (2θ > 57°) are very broad, especially for T = 80 and 100 °C. The values of strains exceeding 1, basically mean that normally distributed interplanar distances d have deviations Δd from the mean value dm higher than the mean interplanar distance itself Δd > dm, which is unreal. In addition, one can see that the 200 peak of the cubic phase and 102 peak of the hexagonal phase presented in the calculated XRD patterns are much narrower relative to the observed in the experimental X-ray diffraction patterns. Due to poor pattern fitting, the reliability factors R are high enough (~20%). Thus, the description of the phase composition by cubic zinc blende and hexagonal wurtzite modifications of Cd_0.7_Mn_0.3_S existing simultaneously is very formal. The peaks of β-Mn_3_O_4_ are very small relative to peaks of Cd_0.7_Mn_0.3_S. The formal quantitative Rietveld analysis gives a ratio of the total weight of two modifications of Cd_0.7_Mn_0.3_S (x~0.3) to the weight of β-Mn_3_O_4_ equal to 0.95:0.05.

### 3.2. Debye Calculation of XRD Patterns Based on the Models on Spherical Nanoparticles of Different Sizes Containing SFs

It can be assumed that XRD patterns correspond to the statistically homogeneous Cd_0.7_Mn_0.3_S particles containing lamellar intergrowth of the ZB and WZ structures. To test this hypothesis, the Debye simulations were carried out. The ZB–WZ type intergrowth can be modeled with the use of SFs in the WZ or ZB structure. Calculated XRD patterns for nanoparticles containing different probabilities of SFs in the WZ structure (α) or in the ZB structure (1 − α) are shown in Figure 4 for spherical particles with diameters (D) of 8 nm and 4 nm. One can see that an increase in the probability of SFs in the WZ structure α leads to the broadening of the 101, 102, and 103 peaks. At α = 0.3 ÷ 0.4, the 102 and 103 peaks look like a halo of diffuse scattering. At α > 0.5, these peaks are not observed in the simulated XRD patterns. At α = 0.2 ÷ 0.5, for relatively large particles (D = 8 nm), the 100 and 101 peaks look like the shoulders of the 002 peak in the WZ structure. However, for relatively small particles (D = 4 nm), the shoulders are not observed in this range of α. Therefore, these XRD patterns could be incorrectly interpreted as being from the ZB structure. One can see that even relatively small particles (D = 4 nm) give the 200 peak in the XRD patterns of the cubic ZB modification (α = 1, 1 − α = 0).

Assuming a two-phase system, some authors use the intensity ratio of the 103 and 110 peaks of the WZ structure to infer the phase ratio ZB:WZ [46,47]. The intensity of 103 peak at ~48° is only assigned to the WZ modification of CdS, while the intensity of the 110 peak at ~44° may be contributed by both ZB and WZ crystal structures. Therefore, for pure ZB and WZ modifications this ratio is ideally zero and nearly one, respectively. However, one can see (Figure 4) that in the case of lamellar intergrowth of ZB and WZ structures in nanoparticles (D < 8 nm) 103 peak is not observed for α = 0.5 ÷ 1 due to the large broadening of this peak. So the intensity ratio of the 103 and 110 peaks is zero not only for pure ZB modification but for the lamellar intergrowth with 0 < WZ:ZB < 1 (0 < (1 − α):α < 1).

### 3.3. Optimized Models of (Cd,Mn)S Nanoparticles

Figure 5 shows the calculated XRD patterns for the optimized models in comparison with the experimental ones for Cd_0.7_Mn_0.3_S obtained by hydrothermal synthesis at temperatures of 80, 100, 120, and 140 °C. One can see that a good agreement with the experimental data was obtained for all temperatures. The results of optimization are summarized in Table 1.

As a result of the model optimization, it turned out that for all temperatures, nanocrystalline particles Cd_0.7_Mn_0.3_S have an ellipsoidal shape, as was the case for CdS particles synthesized under the same conditions [44]. Ellipsoids are elongated along the direction normal to the planes of SFs. The average radii of ellipsoidal particles increase with the temperature (Figure 6). If we compare the dependence of radii on temperature for Cd_0.7_Mn_0.3_S and CdS nanoparticles, we can see that initially (T = 80 °C), the average size of Cd_0.7_Mn_0.3_S particles is larger than the size of CdS particles. However, as the temperature increases, the size of Cd_0.7_Mn_0.3_S particles increases more slowly compared to CdS. It can be assumed that the nanocrystalline β-Mn_3_O_4_ located between Cd_0.7_Mn_0.3_S particles slows down their growth.

A comparison of the defect structures of Cd_0.7_Mn_0.3_S and CdS nanoparticles synthesized at T = 80 °C and 100 °C shows that both compounds have a very disordered structure (Figure 7). However, the Cd_0.7_Mn_0.3_S structure is more WZ-like (α = 0.47, 1 − α = 0.53), and the CdS structure is more ZB-like (α = 0.55, 1 − α = 0.45). Further increase in the processing temperature leads to a decrease in the probability of SFs in the WZ structure α, from 0.47 to 0.36 for Cd_0.7_Mn_0.3_S and from 0.55 to 0.46 for CdS. Therefore, with increasing temperature, the structure of both compounds transforms in the direction from ZB to WZ structure, and Mn^2+^ ions facilitate this transformation.

TEM study of the Mn-containing sample prepared by HTS at 100 °C confirms our calculations (Figure 8). One can see that the particle shape is ellipsoidal, and the particles contain SFs. The particles are overlapped on the TEM image, so it is not possible to calculate the average radius of ellipsoids. We outlined the borders for only four ellipses with different sizes of 11.4 nm:8.2 nm, 10.0 nm:7.3 nm, 9.5 nm:7.3 nm, and 7.8 nm:6.2 nm. These sizes are close to the average values 2 × Rc = 11 nm and 2 × Rab = 7 nm.

### 3.4. Photocatalytic Activity

The activities of the samples of the two series were investigated in the photocatalytic evolution of hydrogen using aqueous solutions containing sodium sulfide and sulfite under visible light irradiation with a wavelength of 450 nm (Figure 9). Cd_0.7_Mn_0.3_S-based catalysts show considerably higher activity relative to pristine CdS, which can be related to the more negative potential of the conduction band of Cd_0.7_Mn_0.3_S in comparison to CdS. The rate of hydrogen evolution for CdS samples synthesized at temperatures of 80, 100, and 120 °C is approximately the same (~0.03–0.04 μmol/min) and approximately doubles for the sample obtained at T = 140 °C (~0.07 μmol/min). The dependence of the hydrogen evolution rate for Cd_0.7_Mn_0.3_S-based photocatalysts on the HTS temperature is more complex. Let us consider factors, which can influence the activity: (1) phase composition; (2) ratio of zinc blende to wurtzite structure; (3) average particle sizes. (1) The phase composition for the Mn-containing series is approximately the same. Along with the main phase Cd_0.7_Mn_0.3_S, there is also a small amount (~5 mass.%) of β-Mn_3_O_4_. (2) As for the structure of mixed sulfides, it depends on the synthesis temperature. With increasing temperature, the structure of (Cd_0.7_Mn_0.3_S)S nanoparticles transforms from almost random polytype (ZB:WZ = 47:53) to, preferably, the WZ structure (ZB:WZ = 36:64) (Figure 7). However, the activity is not directly correlated with the structure. (3) The average particle sizes gradually increase with the temperature. On the one hand, an increase in the particle size, as a rule, leads to a decrease in the specific surface area that reduces the activity of the catalysts. On the other hand, an increase in particle size leads to a decrease in the band gap energy [48], which, in turn, increases the activity of photocatalysts. Apparently, the superposition of these two differently directed dependences can lead to the appearance of a maximum (T = 100 °C) and a minimum (T = 120 °C) on the curve of dependence of the hydrogen evolution rate on temperature.

## 4. Conclusions

Composite photocatalysts for hydrogen evolution were obtained by coprecipitation of Cd^2+^ and Mn^2+^ ions (Mn:Cd = 0.4:0.6) followed by hydrothermal treatment at temperatures of 80, 100, 120, and 140 °C. They contain Cd_0.7_Mn_0.3_S as the main phase (~95 mass%) and a small amount of β-Mn_3_O_4_ (~5 mass%). Commonly applied Rietveld analysis carried out for the zinc blende and wurtzite modifications, showed bad fitting of experimental XRD patterns. The Debye Function Analysis showed that nanosized Cd_0.7_Mn_0.3_S particles have a very disordered structure. In addition, it was shown that the particle shape is ellipsoidal, and the ellipsoids are elongated along the direction perpendicular to the plane of stacking faults. An increase in the hydrothermal synthesis temperature leads to an enlargement of the particles and a gradual decrease in the probability of SFs from 0.47 to 0.36 in the WZ structure. Therefore, with increasing temperature, the structure of Cd_0.7_Mn_0.3_S nanoparticles transforms from almost random polytype (ZB:WZ = 47:53) to, preferably, the WZ structure (ZB:WZ = 36:64). Mn^2+^ ions facilitate the phase transformation in the direction from zinc blende to wurtzite through the formation of disordered structures. Unfortunately, no correlations have been found between hydrothermal synthesis temperature and photocatalytic activity. In this case, one can make an unambiguous conclusion that the activity of (Cd_0.7_Mn_0.3_)S-based photocatalysts in hydrogen production under visible light is much higher than that of pristine CdS.

## Figures and Tables

**Figure 1 materials-16-00692-f001:**
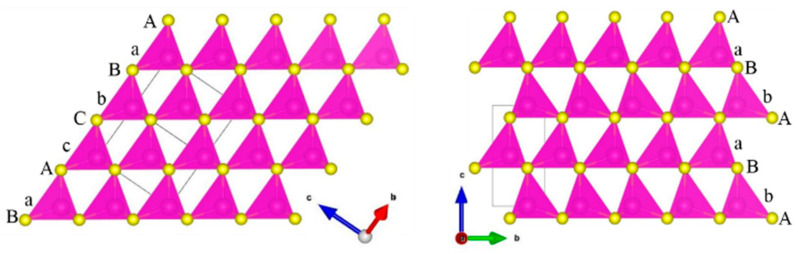
Sketch of ZB (**left**) and WZ (**right**) structures: different stacking of layers of tetrahedrons along the [111] (cubic) and the [001] (hexagonal) axes. Capital and lower case letters designate anion and cation positions respectively.

**Figure 2 materials-16-00692-f002:**
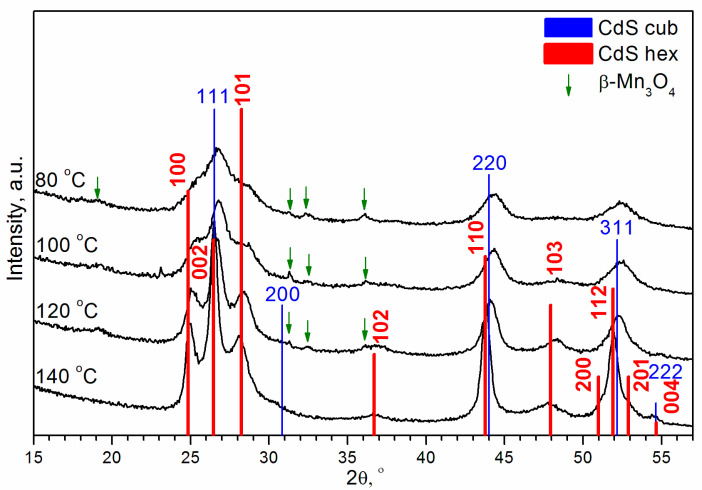
Experimental XRD patterns of (Cd,Mn)S-based photocatalysts prepared by hydrothermal synthesis (HTS) at different temperatures.

**Figure 3 materials-16-00692-f003:**
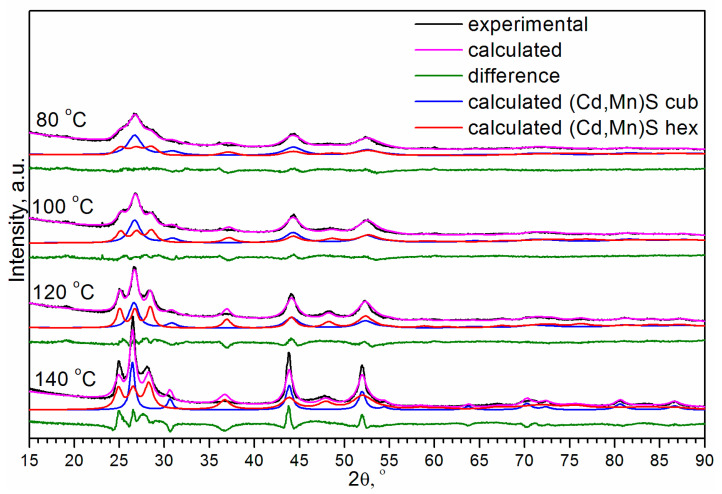
Rietveld analysis of experimental XRD patterns of (Cd,Mn)S as main compound of samples prepared by HTS at different temperatures.

**Figure 4 materials-16-00692-f004:**
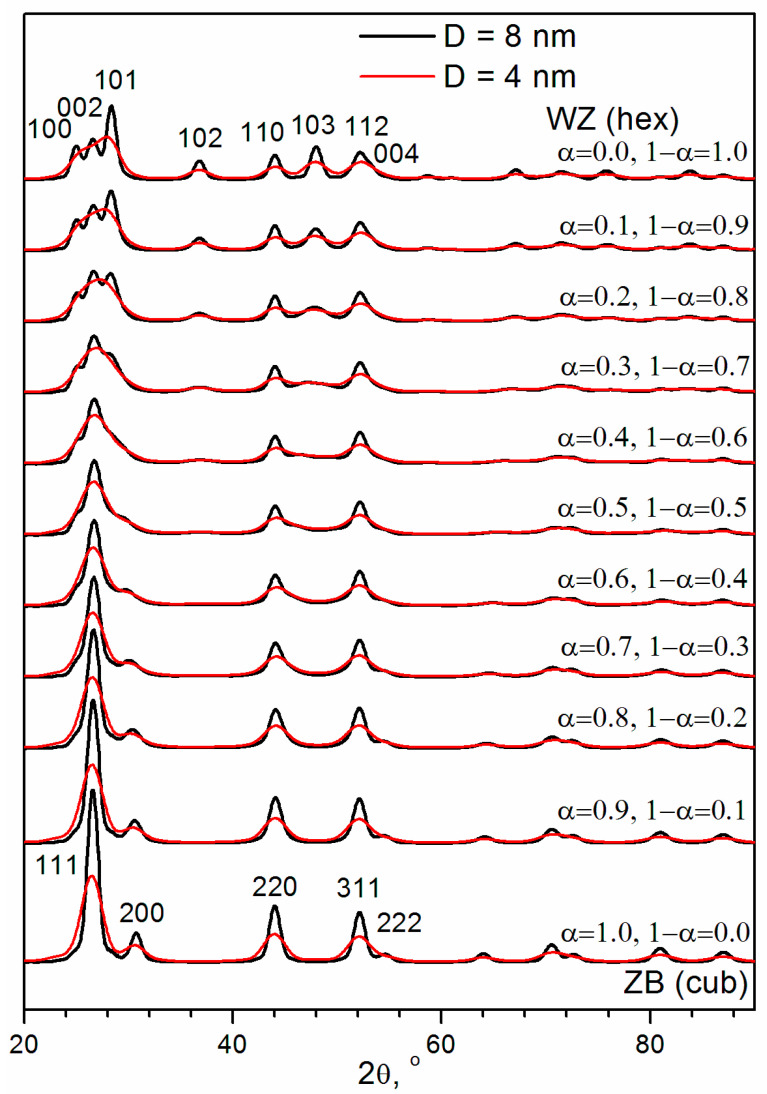
Debye simulation of XRD patterns for spherical CdS nanoparticles having different diameters D and containing different probability of SFs in WZ structure (α) or in ZB structure (1 − α).

**Figure 5 materials-16-00692-f005:**
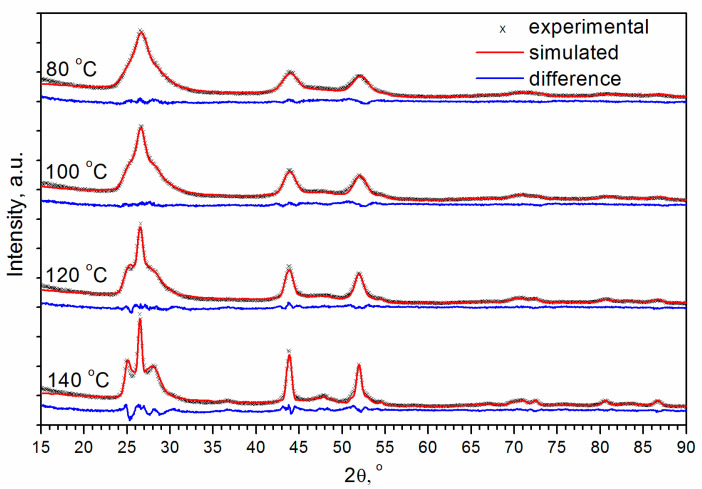
Debye-simulated XRD patterns for Cd_0.7_Mn_0.3_S nanoparticles of ellipsoidal shape containing SFs in comparison with experimental data. Optimized parameters are lattice constants, radii of ellipsoids, and probability of SFs.

**Figure 6 materials-16-00692-f006:**
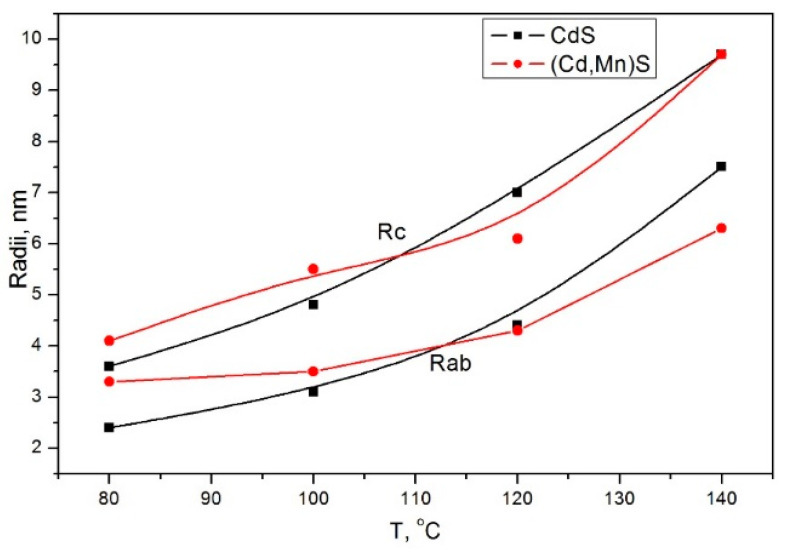
Dependence of radii Rab (in ab plane) and Rc (along c direction) of ellipsoidal particles on temperature T.

**Figure 7 materials-16-00692-f007:**
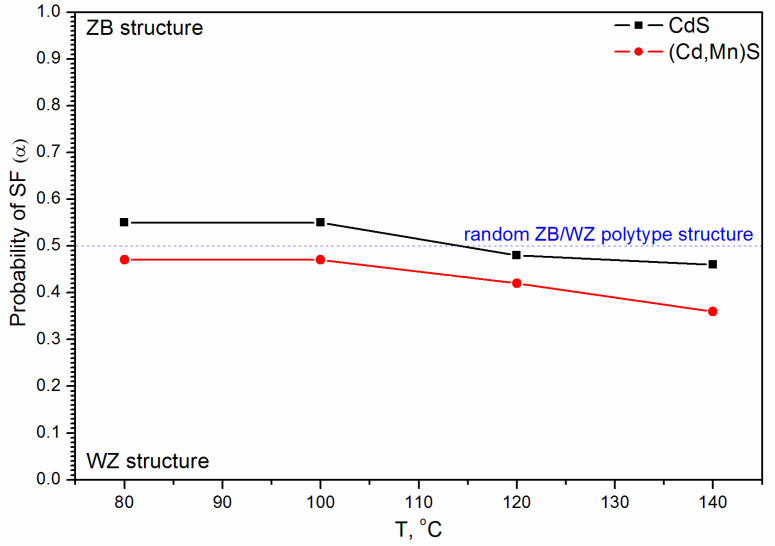
Dependence of probability of SFs in WZ structure (α) on temperature T.

**Figure 8 materials-16-00692-f008:**
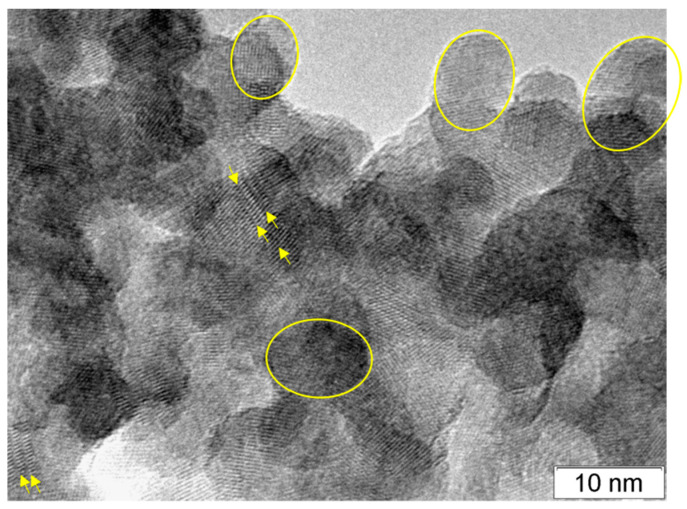
TEM image of Cd_0.7_Mn_0.3_S nanoparticles (HTS at T = 100 °C). The yellow ellipses show particle shape, and the yellow arrows show SFs.

**Figure 9 materials-16-00692-f009:**
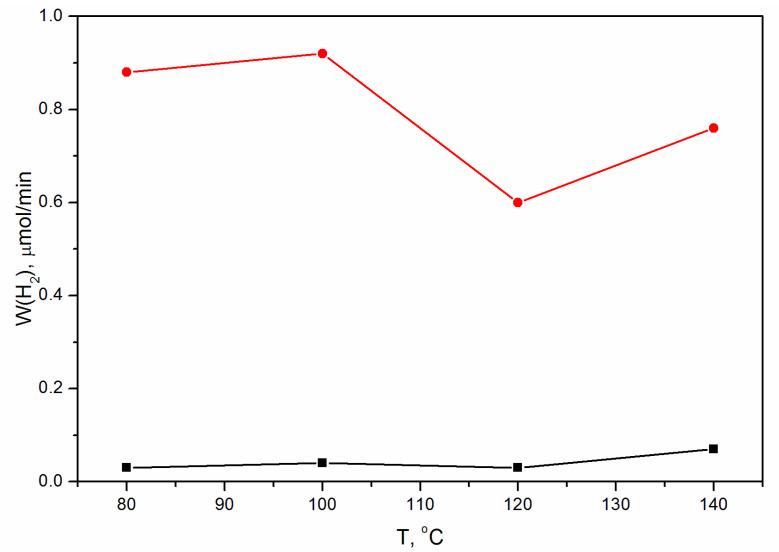
Dependence of the hydrogen evolution rate for CdS (black squares) and Cd_0.7_Mn_0.3_S-based catalysts (red circles) on the synthesis temperature T.

**Table 1 materials-16-00692-t001:** Optimized values of probability of SFs in WZ structure (α), radii of ellipsoidal particles in ab plane (Rab) and in c direction (Rc) in dependence on temperature T.

T, °C	α	Rab, nm	Rc, nm	R, %
80	0.47	3.2	4.1	6.6
100	0.47	3.5	5.5	6.4
120	0.42	4.3	6.1	7.4
140	0.36	6.3	9.7	9.6

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
