# Peer review of "(Cd,Mn)S in the Composite Photocatalyst: Zinc Blende and Wurtzite Particles or Integrowth of These Two Modifications?"

_materials, 2023, doi:10.3390/ma16020692_

Round 1

Reviewer 1 Report

The proposed paper is written very clearly and in good English. The overall merit and quality are very high. I would recommend some minor corrections to clarify the text further. Also, some minor editorial corrections would be beneficial:

Figure 1, along with the description in the text, could be switched. Since we are used to reading left-to-right, the description (and the image) following the same rule would be more practical,

The superscripts are sometimes shown incorrectly – table 1 and lines: 179, 193, 245, 254-255

The formatting in the references should be adjusted. For example, the title in line 301 is written using capital letters, or an issue with the subscripts like in (among others) line 297 (<inf>) should be fixed.

Reviewer 2 Report

Comments on the manuscript entitled “Mn-doped CdS photocatalysts for hydrogen evolution: Debye Function Analysis of the disordered structure and particle shape” Cherepanova et al.
After reading this article, I think it is currently not suite for this special issue on photocatalytic materials due to some points:

1)      This study focused on the phase analysis based on XRD data of Mn-doped CdS materials, however, the title used the attractive word “photocatalyst for hydrogen evolution reaction”, in fact, there was no measurement for photocatalytic performance here. Therefore, the author should provide some photocatalytic measurements, or in order to avoid misleading the title, the author should remove unrelated words from the title.

2)      Some typos should be carefully checked out

3)      Author used the term “CdS-MnS photocatalyst”, but the resulting XRD includes Mn3O4 peaks, indicating the formation of these oxides, thus, this term is not suitable for this article.

4)      The XRD patterns show that samples include CdS cubic, CdS hexagonal, and Mn3O4, therefore, the Rietveld method should be used with these three phases. Did the author do that?

5)      What is the purpose of this research? The author calculated the lattice parameters however, they are not used to determine or support any explanation of photocatalytic properties.

Round 2

Reviewer 2 Report

After revision, the manuscript was much improved. Therefore, It can be accepted